# Antecedent Soil Moisture Conditions Influenced Vertical Dust Flux: A Case Study in Iran Using WRF-Chem Model

**Farshad Soleimani Sardoo** [1], **Tayyebeh Mesbahzadeh** [2,*], **Ali Salajeghe** [2], **Gholamreza Zehtabian** [2], **Abbas Ranjbar** [3], **Mario Marcello Miglietta** [4] **and Nir Krakauer** [5]

[1] Department of Ecological Engineering, Faculty of Natural Resources, University of Jiroft, Kerman 76169-14111, Iran; f.soleimani@ujiroft.ac.ir

[2] Department of Reclamation of Dry and Mountainous Regions, Faculty of Natural Resources, University of Tehran, Tehran 14179-35840, Iran; salajegh@ut.ac.ir (A.S.); g.zehtab@ut.ac.ir (G.Z.)

[3] Atmospheric Science & Meteorological Research Center, Tehran 14179-35840, Iran; a-ranjbar@irimo.ir

[4] National Research Council of Italy-Institute of Atmospheric Sciences and Climate (CNR-ISAC), Corso Stati Uniti 4, 35127 Padua, Italy; m.miglietta@isac.cnr.it

[5] Department of Civil Engineering, The City College of New York, New York, NY 10031, USA; nirkrakauer@gmail.com

\* Correspondence: tmesbah@ut.ac.ir

**Abstract:** Soil moisture is one of the most important parameters affecting dust emission flux. This study was conducted to investigate the effects of soil moisture on vertical dust flux in the central plateau region of Iran. In this study, the WRF-Chem (Weather Research and Forecast with Chemistry) model, with the GOCART (Global Ozone Chemistry Aerosol Radiation and Transport) scheme, was used to estimate the dust emission flux during a major storm from 19 to 21 July 2015, and to discriminate between dust sources. The results showed that the Kyrgyz deserts in Turkmenistan, the Arabian deserts in Saudi Arabia, the deserts of Iraq, and the Helmand region in Afghanistan are sources of foreign dust. Additionally, the central desert plain was identified as an internal dust source, where the dust level reached 7000 $\mu g\,m^{-2}\,s^{-1}$. The results of WRF-Chem simulation were verified with reanalysis data from MERRA2 and AERONET data from Natanz station, which showed good agreement with the simulation. Based on the GLDAS reanalysis, soil moisture content varied between 2.6% and 34%. Linear and nonlinear regression of vertical dust flux values and soil moisture showed nonlinear behavior following the exponential function, with a correlation coefficient of 0.8 and a strong negative association between soil moisture and vertical dust flux.

**Keywords:** dust emission flux; WRF-Chem; GOCART wind erosion scheme; soil moisture; GLDAS; exponential function

## 1. Introduction

As atmospheric dust increases, air quality decreases and becomes dangerous for human health [1,2]. Furthermore, dust particles, due to the absorption and scattering of long-wave and short-wave radiation, greatly affect the radiation budget of the atmosphere and land surface, and thus influence climate [3]. In recent years, scientific communities have been conducting extensive studies to model dust particles rising from the surfaces of deserts under windy conditions and entering the atmosphere [4–6]. The amount of dust rising from the surface of the desert is estimated at between 400 and 4500 Tg/year [7,8]. The dust cycle can be simulated by applying physical–chemical equations that model the transfer, distribution, and conversion of particles. These models have the ability to calculate the particle concentration using information on particle emission rate, emission source characteristics, local topography and regional meteorology. Recent studies have shown that numerical dust prediction models have been improved to better understand the effects of dust particles on the atmosphere. Although numerical simulation results

are often acceptable, uncertainty is observed in the estimated values for dust emission [9]. In recent years, many numerical prediction models have been designed to simulate dust emission and transport [10–20]. These dust emission schemes require different input parameters [10,21,22], depending on how each one parameterizes wind erosion. One of the wind erosion schemes widely used in the prediction of dust particle emissions is the GOCART model [23]. This model includes the components of dust emission, transfer and deposition. With GOCART implemented within the WRF-Chem mesoscale atmospheric chemistry and transport model, the combined system can simulate the transport of particles and their climate feedback in the atmosphere [24–26]. The authors of [27] examined the representation in the WRF-Chem model of dust flux as a function of land surface characteristics to forecast springtime dust concentrations in East Asia, and the results showed that the WRF-Chem model performs well in reproducing horizontal and vertical distributions of optical properties of dust, as compared with satellite data. Ref. [28] reported on the spatial and temporal characteristics of dust events and their participation in the aerosol budget in East Asia. In this study, the WRF-Chem model was used with a new parameterization for dust devils. The results suggested that this source of dust emission peaks in the summer and in early afternoon. Ref. [29] used the WRF-Chem model for the modeling of dust in East Asia during summer 2010. The results showed that the WRF-Chem model could effectively represent the temporal and spatial distribution of dust meteorological factors in summer. Central Iran experiences frequent dust storms, and contains two deserts that are important dust sources. There has been little field study and research on these deserts in terms of wind erosion and dust storms. Generally, particle size in the Loot desert ranges from 0.1 to 0.15 mm, due to the presence of sand dunes. In the inner deserts, sand particles form sandy areas; in the case of sandstorms, sand particles can usually be displaced. Moreover, in deserts, bedrocks, abandoned lands, dried-up wetlands, etc., particles smaller than 65 microns can cause dust storms.

Atmospheric conditions, land surface and soil conditions are the main factors influencing dust emission [30]. The separation of soil particles from the land surface and their emission depends on the threshold friction velocity. In other words, the minimum wind velocity must be above the threshold friction velocity so that the wind can move the particles on the soil surface [31–33]. Threshold velocity depends on soil surface components, such as soil texture [34], soil surface moisture [35], vegetation, snow cover, and soil crust [36,37]. In wind erosion models, dust emission rate is highly dependent on the threshold velocity of soil erosion [38,39]. The most important factor affecting threshold velocity of wind erosion is soil moisture. The amount of soil erosion thus depends on the water resources [40]. Moreover, one of the most important obstacles to economic development in arid and semi-arid regions is the lack of water resources [41–43]. Water resources in arid and semi-arid regions are very rare and there is a poor understanding of them. Water budgets can be assessed on the smallest scale with accurate in situ measurements and at large scale with modern remote sensing and modeling methods [44–46]. The presence of water, particularly in the form of soil moisture, can increase wind erosion threshold velocity, as well as drastically reduce dust emissions. In various studies in experimental environments and wind tunnels, the relationship between soil moisture and dust emission flux has been investigated [47,48]. Ref. [49] showed that, with increasing soil moisture the wind erosion threshold velocity increases and, as a result, the dust emission flux decreases. However, it is very difficult to determine the spatial pattern of soil moisture on a large scale. The use of Global Land Data Assimilation System (GLDAS) re-analysis data is very suitable for estimating the distribution of soil moisture on a large scale. GLDAS is a project focused on the possibility of estimating the components of water balance, especially soil moisture, across the world, using available observed and modeled climate and land surface property data [50].

Given the above considerations, the aim of the present study was to investigate the relationship between soil moisture and dust emission flux in an arid dust source region. The present study used the WRF-Chem model and the GOCART wind erosion scheme to simulate the storm of 19–21 July 2015 over Iran, and to analyze the modeled distribution

of dust emission flux and its relation to soil moisture. The GLDAS re-analysis database was used to calculate the soil moisture. This builds on a previous study by our team [51] that validated the ability of WRF-Chem combined with GOCART to simulate emission flux during other intense dust storms in Iran.

## 2. Materials and Methods

### 2.1. Study Area

Iran's Central Plateau is an extensive hyper-arid basin, with very low precipitation and high potential evaporation [52]. Relative humidity is usually low, and temperature seasonality is intense [53]. Dust storms are accordingly a relatively common problem in this region. Figure 1 shows synoptic stations in the region that observe horizontal visibility. Severe dust storms are defined as those that reduce visibility to under 1000 m.

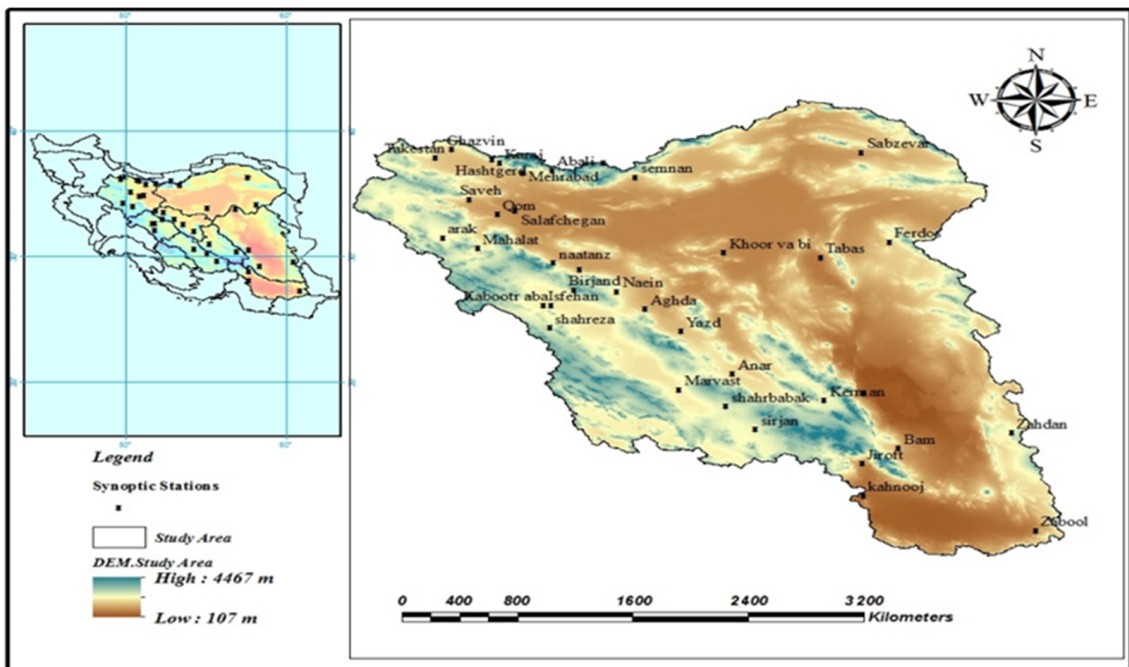

**Figure 1.** Topography and weather stations in Iran's Central Basin region.

### 2.2. WRF-Chem Model and GOCART Dust Emission Scheme

The model configuration is basically the same as that previously used to simulate dust storms in Iran [51]. Briefly, the Weather Research and Forecasting model with Chemistry (WRF-Chem; https://ruc.noaa.gov/wrf/wrf-chem/, accessed on 1 January 2020), version 3.9.1, is a nonhydrostatic limited area numerical model, able to simulate meteorological fields and dust particle transport and their reciprocal feedbacks.

The GOCART scheme uses the following empirical relationship for dust emission rate [54].

$$F_p = C_G S s_p U_{10}^2 (U_{10} - U_t), \ U_{10} > U_t$$

where $C_G$ is a dimensionality coefficient, $s_p$ is the fraction of each size class, $S$ is an erodibility function based on local elevation, $U_t$ is the threshold velocity, and $U_{10}$ is horizontal wind speed at 10 m height. According to [29], the $s_p$ values are 0.1 for particle diameters of 0.1–1 µm, and 1.3 for the 1–1.8 µm, 1.8–3, and 3–6 µm. The threshold velocity is based on [55] as implemented by [21] in the WRF-Chem model.

Model domains, parametrization, initial and boundary conditions used the same settings and input data as previously [51] (Table 1). Also, The model was run at a distance of 27 km. To simulate the investigated storms, the model was run starting 24 h before the storm, and the first 3 h of the run were discarded as spin-up time.

**Table 1.** Schemes used in WRF-Chem implementation.

| WRF Single-Moment 5-Class | Micro-Scale Physics |
|---|---|
| RRTM (Mlawer, 1997) | Long wave radiation |
| Goddard shortwave (Chou, 1998) | Shortwave radiation |
| Noah Land Surface Model (Chen, 1996) | Surface Physics |
| YSU (Noh et al., 2002) | Boundary layer |
| Grell 3D (Grell, 1993) | Cumulus Convection |

*2.3. GLDAS Reanalysis Database Soil Moisture Data*

Soil moisture is the moisture content of the soil, which is affected by the components of the hydrological cycle (precipitation, evaporation, transpiration, etc.). The soil moisture is a source of water for plant growth, and it also promotes soil particle retention (by creating adhesion force). In this study, the GLDAS reanalysis database was used to estimate soil moisture. In order to investigate the effect of soil moisture status on dust flux in the study area, the amount of surface-layer moisture at a depth of 0 to 10 cm in the soil was examined using GLDAS-NOAH data with a spatial resolution of 0.25 degree.

The Global Land Data Assimilation System (GLDAS) was developed jointly by scientists from the National Aeronautics and Space Administration (NASA), the Goddard Space Flight Center, the National Oceanic and Atmospheric Administration, and the International Center for Environmental Forecasting, to produce consistent global estimates of land surface states, including the water budget. This model is designed to integrate satellite products and terrestrial observations along with modeled physical relationships for optimal estimates of ground surface flux and water and energy budgets. Currently, the data of this global system are available in two versions, 2.0 and 2.1, for periods of 1948–2010 (observation-based) and 2015–2000 (climate simulation-based). GLDAS uses four land surface models: Mosaic, Noah, CLM, and VIC [56] (Table 2). The data used as input in this model include meteorological information and ground surface conditions. GLDAS output data are available in GRIB format. Table 2 shows some details of the various GLDAS model products.

**Table 2.** General characteristics of GLDAS data.

| Water and Energy Components | Contents |
|---|---|
| 60° S to 90° N | Geographical latitude |
| 180° W to 180° W | Geographical longitude |
| 1°, 0.25° and 0.12° | Spatial Resolution |
| 3 h & monthly mean | Temporal resolution |
| GLDAS 2.0 January 1948–2021 December 2010 | Time cover |
| GLDAS 2.1 1 March 2001 to present | |
| 360 × 150 for data with 1° resolution; 1440 × 600 for data 0.25° resolution | Dimensions |
| CLM 2.0 | Earth surface models |
| MOSAIC | |
| NOAH 2.7.1 | |
| VIC water balance | |

**3. Results**

In order to investigate the effects of spatial–temporal patterns of soil moisture on vertical dust flux using WRF-Chem model and GOCART wind erosion schematics, the first step was to select a severe and widespread dust storm in the study area. Based on this criterion, using the meteorological codes 06 and 07 and the data on horizontal visibility

at the synoptic stations located in the study area, a storm that took place on 19–21 July 2015 was selected. Figure 2 shows a diagram of the horizontal visibility of the synoptic stations affected by the dust storm. As shown in Figure 2, at some stations (Sabzevar, Semnan, Ferdows, Shahroud) the horizontal visibility was less than 500 m due to the high concentration of dust in the atmosphere.

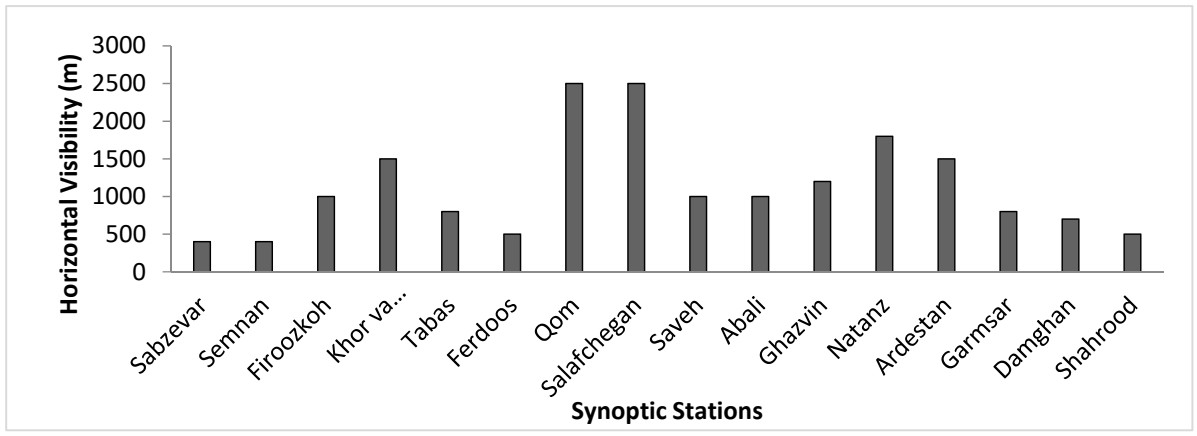

**Figure 2.** Minimum horizontal visibility diagram of some synoptic stations in the study area on 19–21 July 2015.

### 3.1. Spatial and Temporal Pattern of Dust Flux Using GOCART

In order to study the spatial pattern of vertical dust flux using GOCART scheme, atmospheric boundary information (GFS [1] data) as well as land information (vegetation information, soil texture, land use, and soil erodibility) were ingested into the model. The WRF-Chem model simulated the storm of 19–21 July 2015 using the GOCART scheme, and the estimated values of the vertical dust flux were extracted to identify the main sources of dust. Figure 3 shows the spatial and temporal pattern of the dust flux during the storm within a time interval of 3 h.

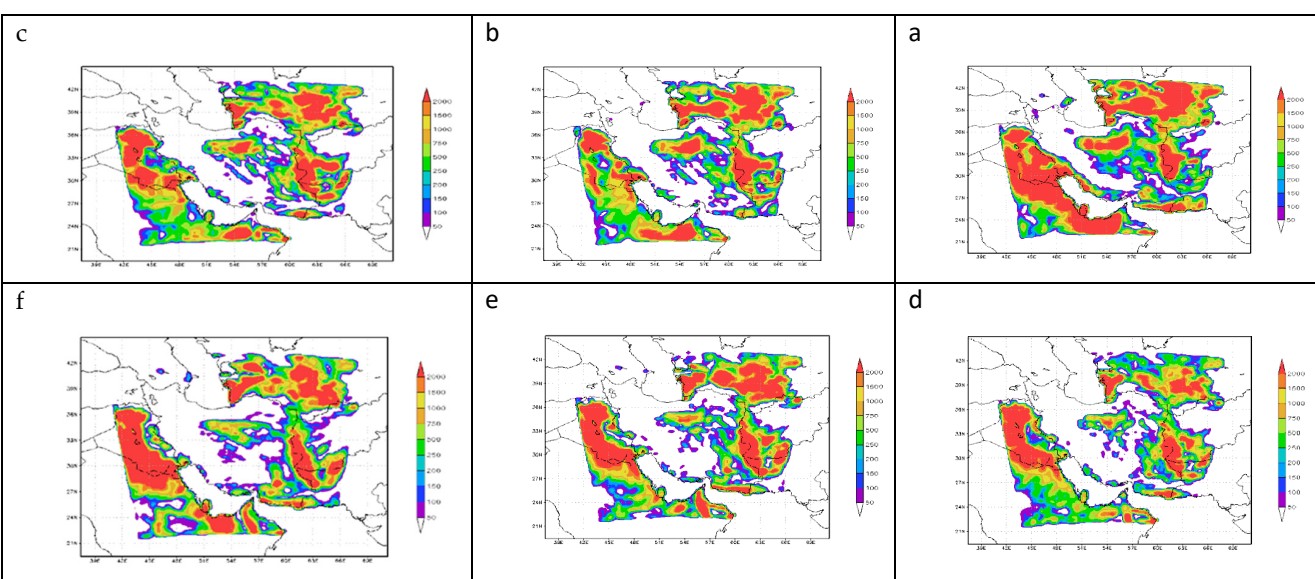

**Figure 3.** *Cont.*

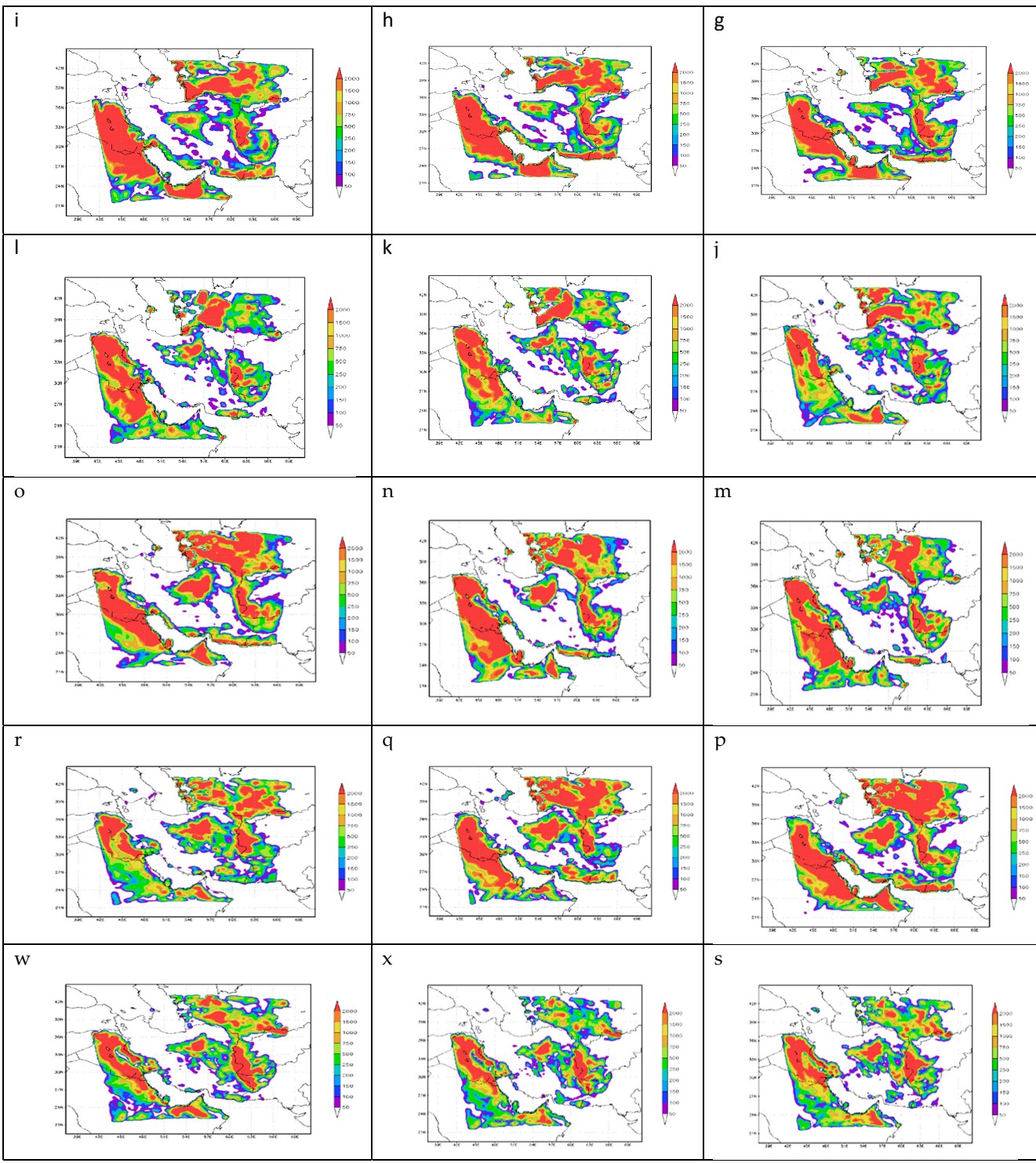

**Figure 3.** *Cont.*

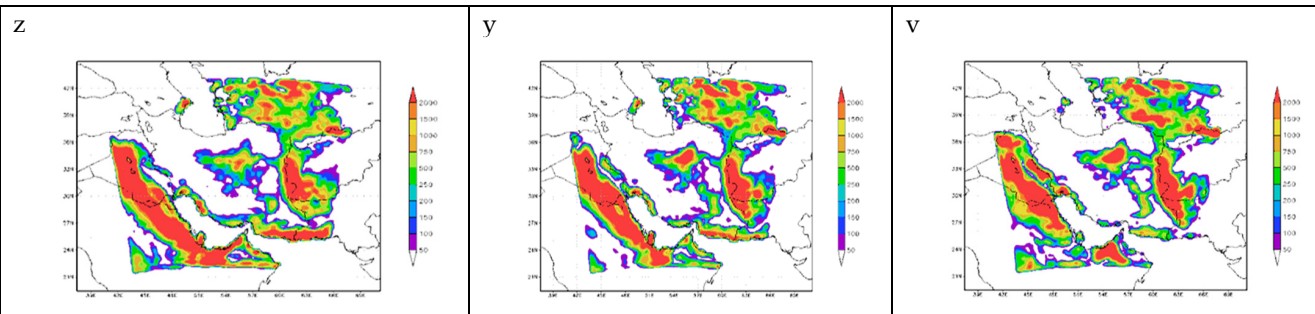

**Figure 3.** Spatial–temporal pattern of simulated vertical dust flux using GOCART scheme and WRF-Chem model $\left(\frac{\mu g}{m^3}\right)$ ((**a**) 18 July 2015 UTC 15:00 (**b**) 18 July 2015 UTC 18:00 (**c**) 18 July 2015 UTC 21:00 (**d**) 19 July 2015 UTC 00:00 (**e**) 19 July 2015 UTC 03:00 (**f**) 19 July 2015 UTC 06:00 (**g**) 19 July 2015 UTC 09:00 (**h**) 19 July 2015 UTC 12:00 (**i**) 19 July 2015 UTC 15:00 (**j**) 19 July 2015 UTC 18:00 (**k**) 19 July 2015 UTC 21:00 (**l**) 20 July 2015 UTC 00:00 (**m**) 20 July 2015 UTC 03:00 (**n**) 20 July 2015 UTC 06:00 (**o**) 20 July 2015 UTC 09:00 (**p**) 20 July 2015 UTC 12:00 (**q**) 20 July 2015 UTC 15:00 (**r**) 20 July 2015 UTC 18:00 (**s**) 20 July 2015 UTC 21:00 (**x**) 21 July 2015 UTC 00:00 (**w**) 21 July 2015 UTC 03:00 (**v**) 21 July 2015 UTC 06:00 (**y**) 21 July 2015 UTC 09:00 (**z**) 21 July 2015 UTC 12:00).

The results show that the outside sources of dust in the western and southwestern parts of Iran are the deserts of Iraq and Saudi Arabia. For the northeastern part of Iran, Ghareghoom desert in Turkmenistan is also known as an outside source of dust. This dust can affect the provinces of North Khorasan, Gorgan, Khorasan Razavi, Semnan, and Qom. The results showed that the amount of dust flux was at a maximum on 20 July 2015 12:00–15:00 UTC. The average simulated spatial pattern of vertical dust flux in the study area based on the simulated storm from 19 to 21 July 2015 shows that the central desert area (Dasht-e Kavir) is the main source of dust in the study area (Figure 4a). The vertical dust flux in this area reached more than 7000 micrograms per square meter per second. If this storm continued for 12 h with a constant intensity at this level, 4302 kg of soil from each hectare of this area would enter the atmosphere, which is a very high amount. Figure 4b shows the main areas affected by the dust phenomenon in central Iran (19 to 21 July 2015). Also, Table 3 shows information on the areas affected by the dust phenomenon in the Central Desert

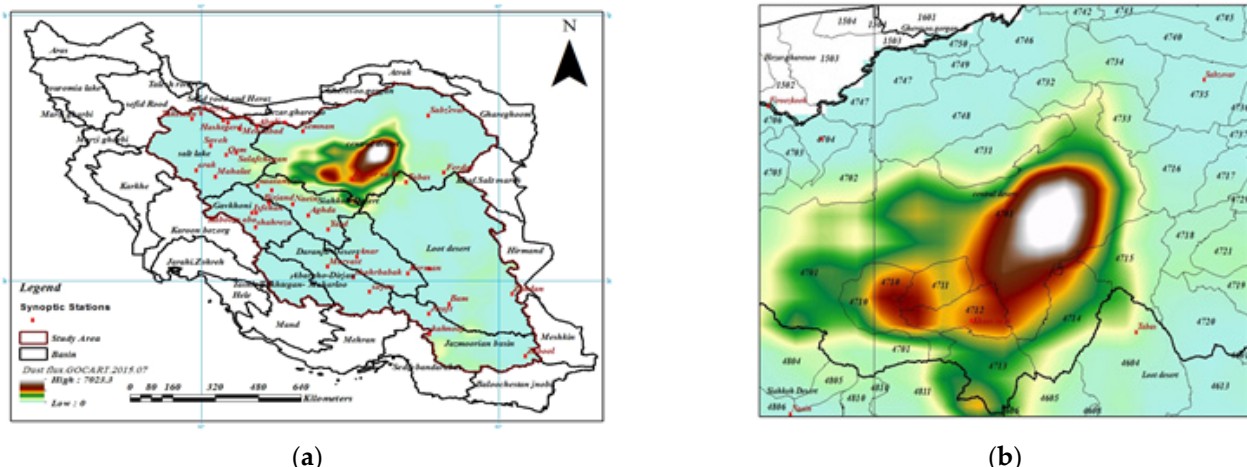

(**a**)                                                                                      (**b**)

**Figure 4.** (**a**) Average spatial pattern of dust flux $\left(\frac{\mu g}{m^3}\right)$ (using WRF-Chem model and GOCART schema (19 to 21 July 2015), (**b**) The main areas affected by the dust $\left(\frac{\mu g}{m^3}\right)$ (phenomenon in central Iran (19 to 21 July 2015).

**Table 3.** Information on areas affected by the dust phenomenon in the Central Desert.

| Description | Code | Description | Code |
|---|---|---|---|
| Rig Zarin Desert | 4811 | Kavir plain | 4701 |
| Semnan Desert | 4702 | Choopanan City | 4710 |
| Terood Area | 4731 | Jandagh City | 4711 |
| Biarjemana Area | 4732 | Khoor-Farokhi Area | 4712 |
| Khartoran Desert | 4733 | Biazeh Area | 4713 |
| Dagh Kavir | 4605 | Robat-Khan Area | 4714 |
| Red Dagh | 4804 | Dastgardan Area | 4715 |

*3.2. Validation of GOCART Scheme Results by Station PM10 Values and MERRA2 Re-Analysis*

In order to verify the results of GOCART scheme, the $PM_{10}$ values of Natanz station located in the study area and also the values of dust surface concentration of the MERRA2 global reanalysis were used. For this purpose, the surface dust concentration values simulated using WRF-Chem combined with the GOCART scheme on 19–21 July 2015 (as shown in Figure 5) were compared with the observed dust concentration values of Natanz station (Figure 6). Also, Figure 7 shows the time series of dust concentration by GOCART scheme related to the storm of 19 to 21 July 2015.

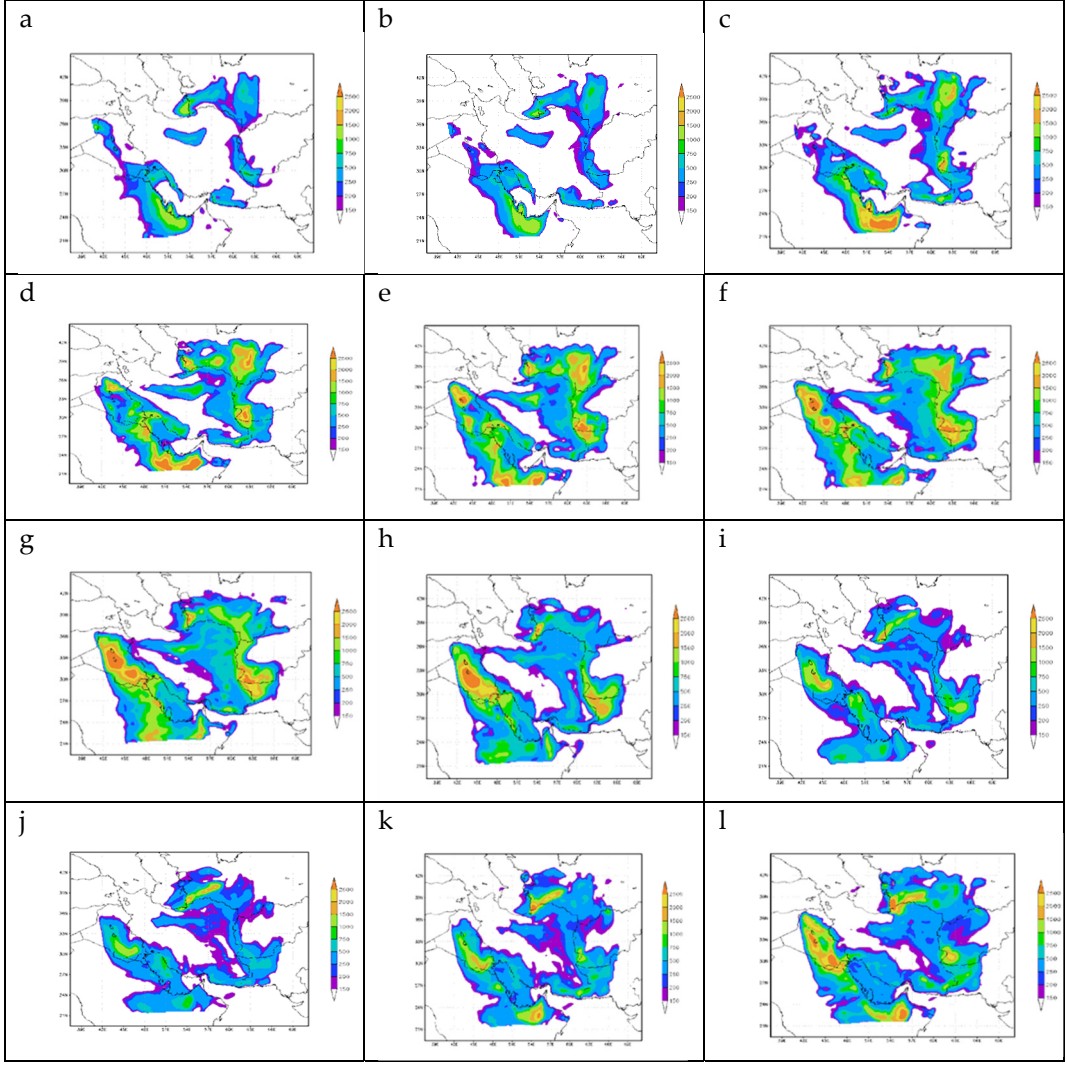

**Figure 5.** *Cont.*

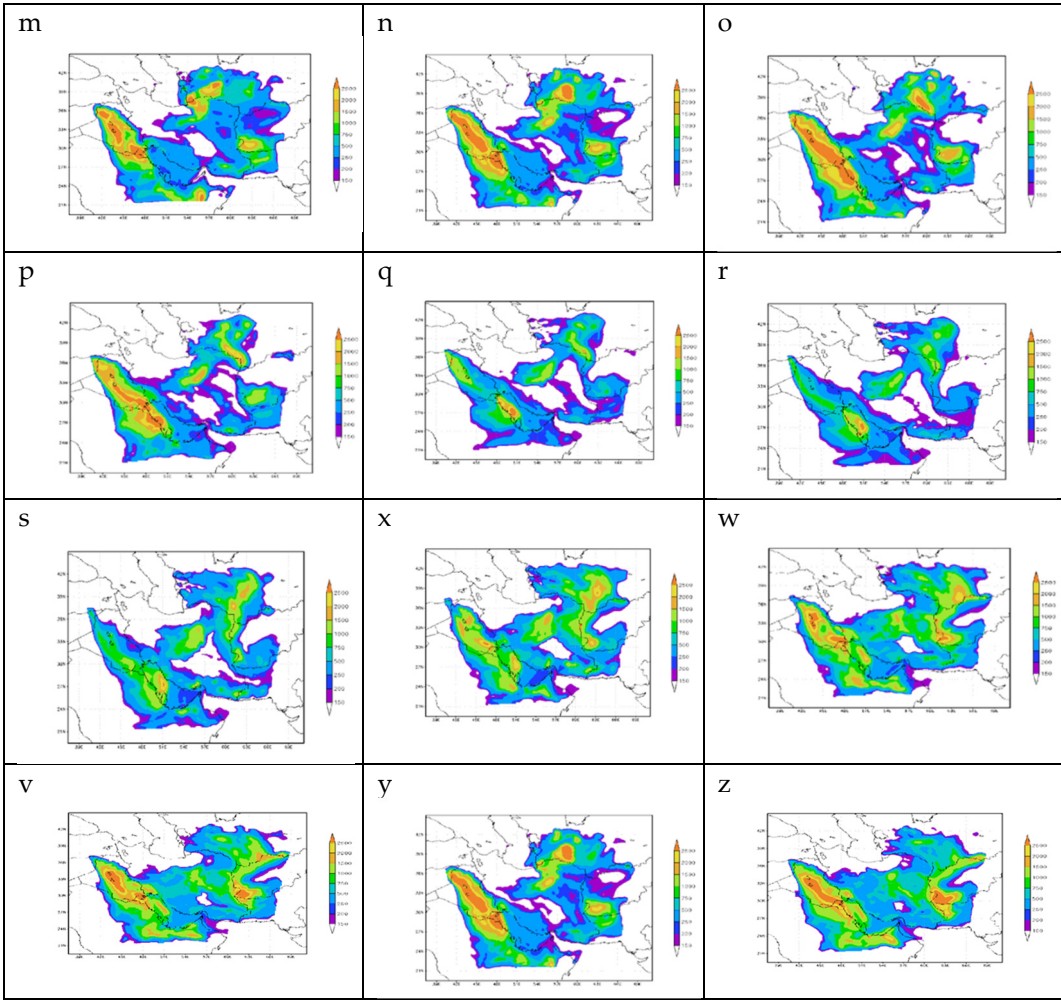

**Figure 5.** Spatial–temporal pattern of simulated dust concentration using GOCART schema and WRF-Chem model $\left(\frac{\mu g}{m^3}\right)$ ((**a**) 18 July 2015 UTC 15:00 (**b**) 18 July 2015 UTC 18:00 (**c**) 18 July 2015 UTC 21:00 (**d**) 19 July 2015 UTC 00:00 (**e**) 19 July 2015 UTC 03:00 (**f**) 19 July 2015 UTC 06:00 (**g**) 19 July 2015 UTC 09:00 (**h**) 19 July 2015 UTC 12:00 (**i**) 19 July 2015 UTC 15:00 (**j**) 19 July 2015 UTC 18:00 (**k**) 19 July 2015 UTC 21:00 (**l**) 20 July 2015 UTC 00:00 (**m**) 20 July 2015 UTC 03:00 (**n**) 20 July 2015 UTC 06:00 (**o**) 20 July 2015 UTC 09:00 (**p**) 20 July 2015 UTC 12:00 (**q**) 20 July 2015 UTC 15:00 (**r**) 20 July 2015 UTC 18:00 (**s**) 20 July 2015 UTC 21:00 (**x**) 21 July 2015 UTC 00:00 (**w**) 21 July 2015 UTC 03:00 (**v**) 21 July 2015 UTC 06:00 (**y**) 21 July 2015 UTC 09:00 (**z**) 21 July 2015 UTC 12:00).

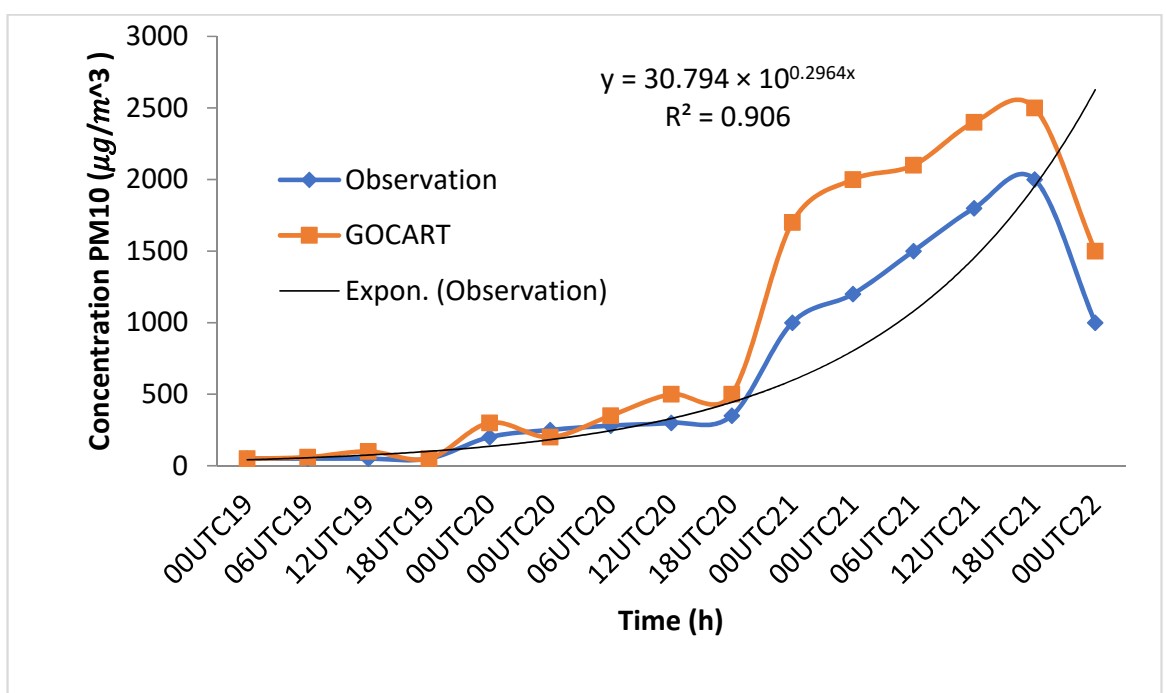

**Figure 6.** PM10 concentration $\left(\frac{\mu g}{m^3}\right)$ at Natanz station using observational data and the outputs from WRF-Chem and GOCART dust emission schemes.

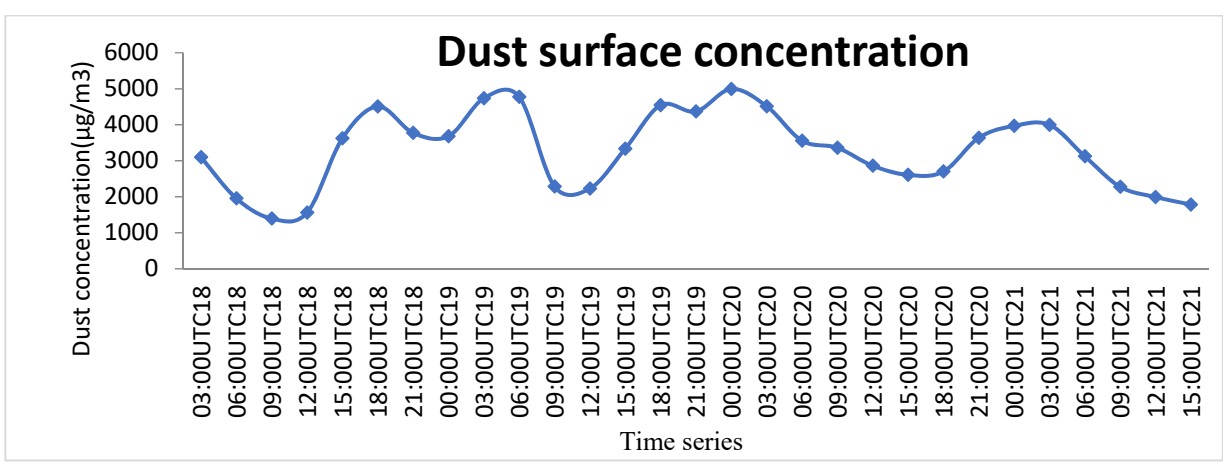

**Figure 7.** Time series of dust concentration by GOCART scheme related to storm of 19 to 21 July 2015.

In order to investigate the temporal pattern of dust surface concentration in the study area, time series of dust surface concentration from the MERRA2 re-analysis with interval 3 h were used (https://gmao.gsfc.nasa.gov, accessed on 1 January 2020).

The results of this study showed that the changes in the dust surface concentration had a sinusoidal pattern, so that from midnight to noon, the dust surface concentration level decreased, and from noon to midnight the dust surface concentration increased. In other words, midnight (00:00 GMT, or early morning local time) tended to have the highest dust surface concentration, and at noon the lowest dust surface concentration was observed. Figure 8 shows the temporal pattern of dust surface concentration using MERRA2 reanalysis database.

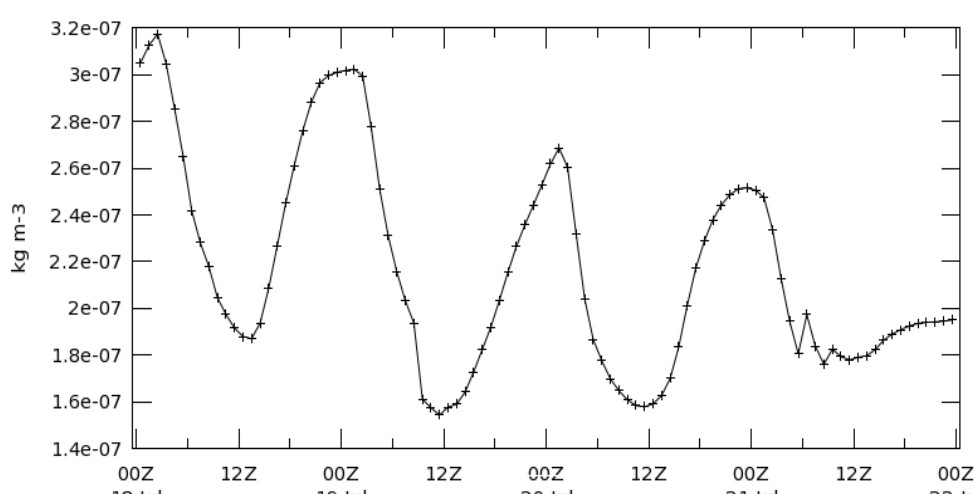

Time Series, Area-Averaged of Dust Surface Mass Concentration, time average hourly 0.5 x 0.625 deg. [MERRA-2 Model M2T1NXAER v5.12.4] kg m-3 over 2015-07-18 00Z - 2015-07-21 23Z, Region 50.1466E, 28.1335N, 59.463E, 36.2194N

- The user-selected region was defined by 50.1466E, 28.1335N, 59.463E, 36.2194N. The data grid also limits the analyzable region to the following bounding points: 50.625E, 28.5N, 59.375E, 36N. This analyzable region indicates the spatial limits of the subsetted granules that went into making this visualization result.

**Figure 8.** The temporal pattern of dust surface concentration using MERRA2 reanalysis database.

### 3.3. GLDAS Soil Moisture

The GLDAS analysis database with spatial resolution of 0.1° was used to study the spatial pattern of soil moisture. The reanalysis data were downloaded from GLDAS in netCDF format and converted to raster format in ArcGIS software to map the soil moisture change pattern in the study area (Figure 9).

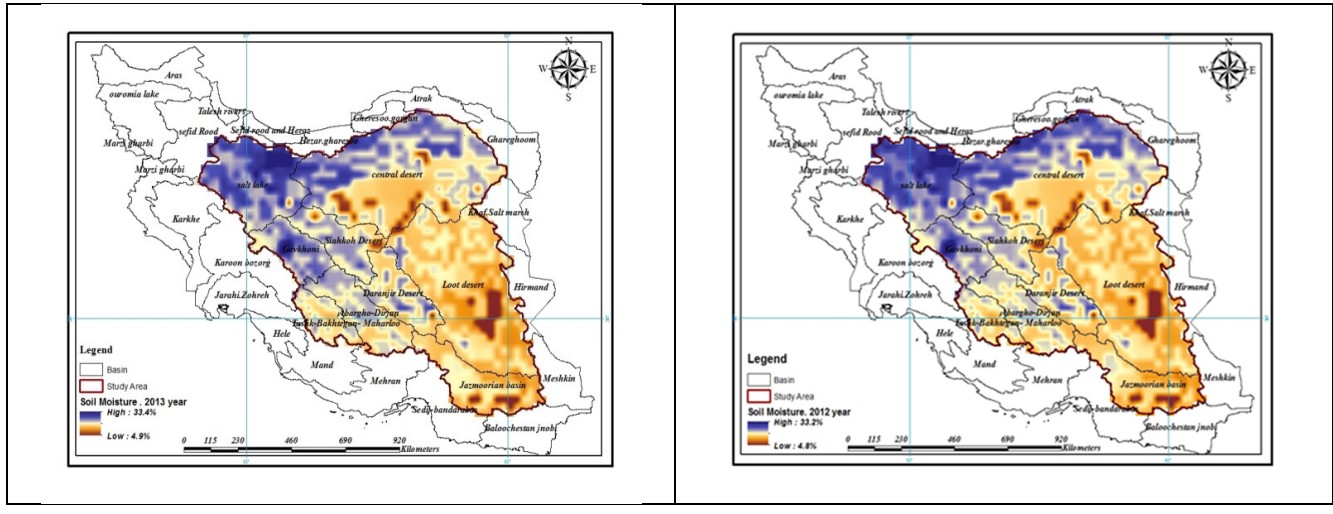

**Figure 9.** *Cont.*

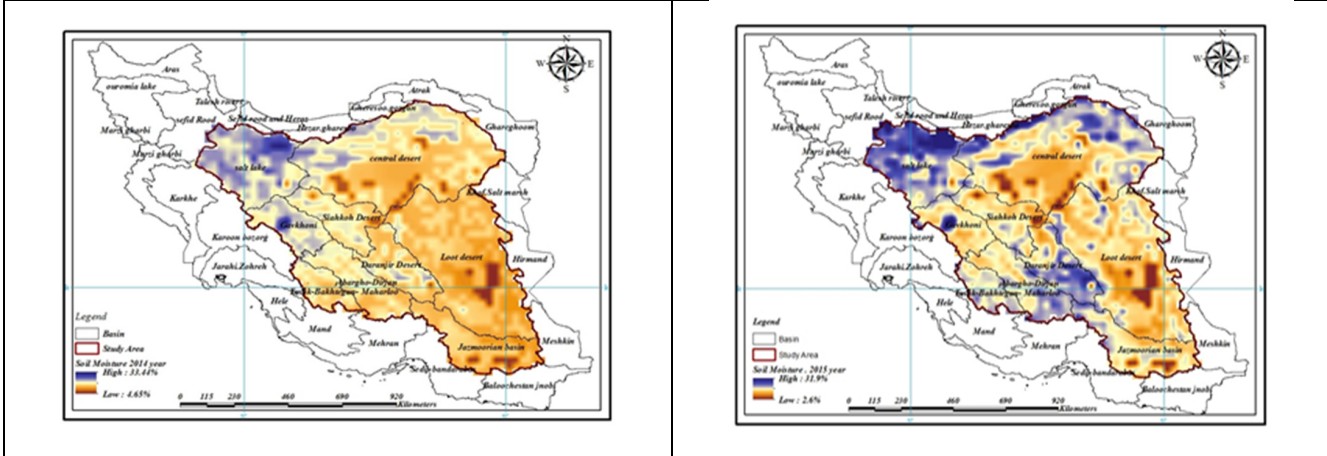

**Figure 9.** The changes in soil moisture at the study area.

The results showed the parts of the Iran central desert in the northeast of the study area, as well as the Loot desert in Kerman, Sistan, and Baluchistan provinces, and the Jazmourian basin in the south had the lowest soil moisture of only about 4 percent. The soil in these areas is expected to be prone to dust and wind erosion

*3.4. The Correlation of Vertical Dust Flux and Soil Moisture*

3.4.1. Correlation Coefficient

GIS and IDRISI software were used to evaluate the correlation between spatial distribution of GOCART vertical dust flux and GLADS soil moisture patterns over the period 2012–2015 in the study area. The results showed that the correlation between soil moisture parameters and vertical dust flux was significant, and their correlation coefficient was slightly different in different years. The highest absolute correlation belonged to 2014 with a coefficient of 0.81 (Table 4).

**Table 4.** Correlation between spatial pattern of dust flux and soil moisture in 2012–2015.

| Correlation Coefficient | Year | Scheme |
|:---:|:---:|:---:|
| 0.79 | 2012 | |
| 0.79 | 2013 | GOCART |
| 0.81 | 2014 | |
| 0.80 | 2015 | |

3.4.2. Linear and Nonlinear Regression

Using the Pearson correlation coefficient, the correlation between soil moisture and vertical dust flux was investigated using SPSS software. Correlation of vertical dust flux with soil moisture was −0.643, which is significant at the 0.01 level.

In contrast with linear regression, nonlinear regression can fit different functional relationships more flexibly between independent and dependent variables. In this study, the highest nonlinear correlation between dust flux and soil moisture was 0.8, reflecting the extent to which dust flux as a dependent parameter can be predicted based on the amount of soil moisture in the study area. Figure 10 shows the best-fit correlation equation between vertical dust flux and soil moisture, where dust flux decreases exponentially with increasing soil moisture level.

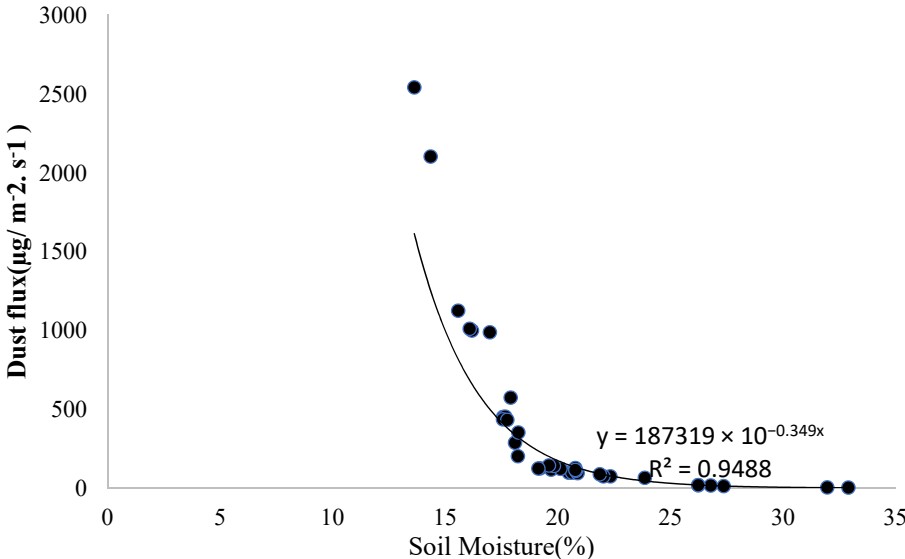

**Figure 10.** Nonlinear regression diagram between soil moisture and vertical dust flux.

## 4. Discussion

Today, the phenomenon of atmospheric dust in arid and semi-arid regions is one of the most important natural hazards. Atmospheric dust causes significant damage to the infrastructure of human societies every year. Identifying dust-producing areas and the factors that affect them is essential for managing and reducing the harmful effects of the phenomenon. In this study, in order to identify dust-generating areas, the modeled dust emission flux was used. The soil-moisture parameter was considered as a major influence on the dust emission flux. The study used the WRF-Chem model and the GOCART wind erosion scheme to simulate the major dust storm of 19–21 July 2015. The GLDAS database was also used to determine the spatial pattern of the water budget, specifically soil moisture, in the study area. The modeling results revealed that the Ghareghoom deserts in Turkmenistan, the Arabian deserts in Saudi Arabia, the deserts of Iraq, and Afghanistan's Hirmand basin are some of the important dust centers surrounding Iran, which can affect the atmosphere of the central plateau of Iran, during this storm. Iran's central desert area (Dasht-e Kavir) was identified as the most important center of internal dust emission. The eastern cities of Isfahan province, especially the Choopanan region and Jandagh city, are at the center of this dust source. The simulation results of WRF-Chem with the GOCART scheme were verified using Natanz station $PM_{10}$ data as well as MERRA2 re-analysis data, which showed the high accuracy of the values simulated by GOCART design. It was this found that the WRF-Chem model paired with the GOCART dust scheme has high efficiency and is a suitable tool for simulation of dust storms in Iran. Similarly, ref. [56] used the WRF-Chem numerical model and satellite data, as well as pollution measuring stations affiliated with the Environment Organization and AERONET stations to investigate the effect of soil storms in the western and southwestern regions of Iran on radiant flux. The results of this study showed that the model's performance in simulating the amount of dust generated during the storm is acceptable, and that the horizontal and vertical distribution of dust simulated by the model and observed by satellite showed similar patterns. This is consistent with the results of the present study.

In order to manage and control dust storms, it is necessary to identify the most important parameters affecting the dust emission flux. Soil moisture is known to be one of the most important parameters affecting dust flux, and it was here estimated using GLDAS. In areas with high dust content, soil moisture was as low as 4%. The results of correlation analysis showed that the spatial pattern of dust flux has a correlation above 0.81 with the spatial pattern of soil moisture. Comparing linear and nonlinear regression showed that dust emission and soil moisture fluctuations have a strong negative correlation and

follow a nonlinear relationship that can be described by an exponential function. Similarly, ref. [57] used observational data and meteorological data to investigate the effects of soil moisture on dust emission flux, which showed a high negative correlation between these two parameters. Ref. [49] examined the effects of soil moisture on sand saltation and the release of dust in China, which showed the driest soils had lower threshold friction velocities for dust emission and could release finer sand particles compared to less dry soils. Ref. [58] examined the relationship between soil moisture and dust diffusion in bare sandy soil in Mongolia, finding a very strong relationship between soil moisture and frictional threshold velocity as well as dust emission, which confirms the results of this study.

In conclusion, our regional simulation of the dust storm that took place over Iran on 19–21 July 2015, supplemented by data from land surface and global climate reanalyses and from station measurements, supports the key role of low soil moisture as a driver of vertical dust flux. This relationship has implications for better prediction of dust storm potential as well as for the management of water resources, which can affect soil moisture and hence dust storm intensity.

**Author Contributions:** All authors contributed to the study conception and design. Material preparation, data collection and analysis were performed by F.S.S., T.M., A.S., G.Z. and A.R. The first draft of the manuscript was written by F.S.S., T.M., M.M.M. and N.K. All authors commented on previous versions of the manuscript. All authors have read and agreed to the published version of the manuscript.

**Funding:** This research received no external funding.

**Institutional Review Board Statement:** Not applicable.

**Informed Consent Statement:** Not applicable.

**Data Availability Statement:** Not applicable.

**Acknowledgments:** We thank Eric Parteli for discussions.

**Conflicts of Interest:** The authors declare no conflict of interest.

## Note

[1]　　GFS: Global Forecast System.

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
