# Peer review of "Antecedent Soil Moisture Conditions Influenced Vertical Dust Flux: A Case Study in Iran Using WRF-Chem Model"

_land, doi:10.3390/land11060819_

Round 1
Reviewer 1 Report
This manuscript applied the WRF-Chem model together with the GOCART scheme to explore the dust emission flux of the storm happened in 19th to 21st July of 2015. I have a concern that I did not see any model validation of WRF-chem model, as well as the meterological conditions (e.g. wind) validation, which have been widely shown in many air quality model studies [1,2,3]. Therefore, suggest a major revision before the manuscript is accepted.
- Line 67: ‘This result’ should be ‘this result’.
- Line 141: Although the model configurations were described in your previous publication, a brief summary of the model setting should be listed here.
- Figure 3: too many figures here. Suggest to show the averaged mean. The time series plots for specific locations should be selected to show the temporal changes instead of the spatial maps.
- Same problems for Figure 6.
- Figure 7: the x-axis is useless and hard to follow, please change.
- Line 263: should be “table 3”
- Table 3 and table 4: Please list the formula of calculating the correlation. Please compare with the correlation used in the model validations in [1, 2]. Surprising why the correlations are negative? Please clarify.
- For all the time series plots, suggest to use the local time instead of the UTC time.
- Line 143, 144, please clarify why the spinning up is only 3 hours? Please clarify the differences of spinning up time between CMAQ model [1, 2] and WRF-chem model.
- What are the resolutions of the WRF-chem model?
[1] Zhang, X., Fung, J. C., Zhang, Y., Lau, A. K., Leung, K. K., & Huang, W. W. (2020). Assessing PM2. 5 emissions in 2020: the impacts of integrated emission control policies in China. Environmental Pollution, 263, 114575.
[2] Hu, J., Chen, J., Ying, Q., & Zhang, H. (2016). One-year simulation of ozone and particulate matter in China using WRF/CMAQ modeling system. Atmospheric Chemistry and Physics, 16(16), 10333-10350.
[3] Yuan, T., Chen, S., Huang, J., Zhang, X., Luo, Y., Ma, X., & Zhang, G. (2019). Sensitivity of simulating a dust storm over Central Asia to different dust schemes using the WRF-Chem model. Atmospheric Environment, 207, 16-29.
Author Response
Reviewer 1
This manuscript applied the WRF-Chem model together with the GOCART scheme to explore the dust emission flux of the storm happened in 19th to 21st July of 2015. I have a concern that I did not see any model validation of WRF-chem model, as well as the meterological conditions (e.g. wind) validation, which have been widely shown in many air quality model studies [1,2,3]. Therefore, suggest a major revision before the manuscript is accepted.
done
- Line 67: ‘This result’ should be ‘this result’. done
- Line 141: Although the model configurations were described in your previous publication, a brief summary of the model setting should be listed here. Done
- Figure 3: too many figures here. Suggest to show the averaged mean. The time series plots for specific locations should be selected to show the temporal changes instead of the spatial maps.
The purpose of these maps is both temporal and spacial dimension of the dust phenomenon so that the reader can better understand the simulation
- .Same problems for Figure 6.
The purpose of these maps is both temporal and spacial dimension of the dust phenomenon so that the reader can better understand the simulation
- Figure 7: the x-axis is useless and hard to follow, please change.
done
- Line 263: should be “table 3” done
- Table 3 and table 4: Please list the formula of calculating the correlation. Please compare with the correlation used in the model validations in [1, 2]. Surprising why the correlations are negative? Please clarify.
Correlations in GIS software are based on spatial maps. You are right. I mistyped. Correlation of Table 5 is linear regression with SPSS software
- For all the time series plots, suggest to use the local time instead of the UTC time.
This is fine, but makes it difficult for readers and users to compare better when comparing MERRA2 data. Because the time data of the MERRA2 database cannot be changed.
- Line 143, 144, please clarify why the spinning up is only 3 hours? Please clarify the differences of spinning up time between CMAQ model [1, 2] and WRF-chem model.
The shorter the time interval, the more changes it is possible to see. The MERRA2 data also had a 6-hour time step, which showed that the changes were both better and easier to compare with the MERRA2 data with the 3-hour time step. And if the time interval was considered less, the number of maps from the beginning of the event to the end of the event would be very large.
- What are the resolutions of the WRF-chem model? 27 km horizontal Resolation

Reviewer 2 Report
The topic of this study is interesting and of great interest for arid and desert regions. Authors have combined different techniques and the manuscript provides an analysis of a specific dust storm. However, there are many aspects that have to be improved/ addressed/ argued, namely:
- I suggest the following title: "Antecedent soil moisture conditions influenced vertical dust flux: A case study in Iran using WRF-Chem model"
- Some affiliations are not complete. Authors should provide the same level of information for all authors. Remove the job description and include the name of the Faculty / Department of all authors.
- Replace "Terrestrial water budget" by "soil moisture" in the whole manuscript.
- Abstract. The sentence "The results of WRF-Chem simulation were verified with reanalysis data of MERRA2 and AERONET data of Natanz station, which shows their high accuracy" is not clear and should be rewritten.
- Abstract. Authors have not included any sentence regarding: I) the research gap; II) the novelty of their findings; and III) the actual contribution to the scientific literature. These three aspects have to be included, at least with one sentence per bullet point.
- Introduction. Please, divide the content into four paragraphs: I) from "Dust phenomenon..." to "...at between 400 and 4500 Tg /yr (Huneeus et al., 2012, Evan et al., 43 2014)."; II) from "The dust cycle can be simulated by..." to "...the aerosol budget in East Asia"; III) from "In this study, WRF-chem model and a parametric..." to "...in different parts of the world over the past 108 few years (Walker & Houser 2004)"; and IV) from "The aim of this study was to investigate..." to the end of the section. After doing this, you should rewrite the content of the four paragraphs because the quality of the text is really poor. Please, present and follow a chain-of-argument in order to have a readable text. Also, there are many typewritten mistakes.
- Study area. I miss more information about the particle size distribution of the materials that lay on the ground in Iran's Central Plateau. As this study is about dust storms, it appears relevant to include this information.
- The term "synoptic station" sounds rare for me. What are the differences between synoptic stations and weather stations?
- Figure 3. The 24 maps have to be presented in one page, with 3 maps per row. It is not necessary to draw the legend in all maps. Besides, authors have not written the units of the legend. The layout of this figure have to be improved. After that, do the same with the 26 maps of Figure 6.
- Figure 4. Add the units of the legend.
- Figure 5 should be presented as Figure 4b because it is a zoom-in view of the map included in Figure 4.
- How the MERRA2 databases were created? From which data? I cannot find the origin of this data. What is the meaning of MERRA2? Some aspects of this study are "dark" and authors should present all information in a "clearer" way.
- Table 4. The content of this table can be presented as part of the text of section 3.4.2., and then, the table can be removed.
- Most content of the Discussion section is a summary of the previous sections. I miss a deep analysis of the results on the basis of the existing literature. Without any doubt, this section has to be split into two sections: Discussion and Conclusions.
- References. The list is long and in some cases authors have not provided any detail/s from these documents. Therefore, I suggest two options: I) reduce the number of references to the most relevant ones; or II) add more information in the text from the cited literature. Please, avoid writing list of references without providing details; that is not valuable.
Author Response
- I suggest the following title: "Antecedent soil moisture conditions influenced vertical dust flux: A case study in Iran using WRF-Chem model" done
- Some affiliations are not complete. Authors should provide the same level of information for all authors. Remove the job description and include the name of the Faculty / Department of all authors. done
- Replace "Terrestrial water budget" by "soil moisture" in the whole manuscript. done
- Abstract. The sentence "The results of WRF-Chem simulation were verified with reanalysis data of MERRA2 and AERONET data of Natanz station, which shows their high accuracy" is not clear and should be rewritten.
done
- Abstract. Authors have not included any sentence regarding: I) the research gap; II) the novelty of their findings; and III) the actual contribution to the scientific literature. These three aspects have to be included, at least with one sentence per bullet point.
done
- Introduction. Please, divide the content into four paragraphs: I) from "Dust phenomenon..." to "...at between 400 and 4500 Tg /yr (Huneeus et al., 2012, Evan et al., 43 2014)."; II) from "The dust cycle can be simulated by..." to "...the aerosol budget in East Asia"; III) from "In this study, WRF-chem model and a parametric..." to "...in different parts of the world over the past 108 few years (Walker & Houser 2004)"; and IV) from "The aim of this study was to investigate..." to the end of the section. After doing this, you should rewrite the content of the four paragraphs because the quality of the text is really poor. Please, present and follow a chain-of-argument in order to have a readable text. Also, there are many typewritten mistakes.
done
- Study area. I miss more information about the particle size distribution of the materials that lay on the ground in Iran's Central Plateau. As this study is about dust storms, it appears relevant to include this information.
There are two deserts in the central plateau of iran Loot and central desert where dust storms are the main problems. Sand dunes are formed in these area. Wind erosion particlese are commonly transported by mutation and suspension. So there is no exact information on the particle size and i did not fond any research on this
- The term "synoptic station" sounds rare for me. What are the differences between synoptic stations and weather stations?
Measurements at synoptic stations are more accurate and more advanced
- Figure 3. The 24 maps have to be presented in one page, with 3 maps per row. It is not necessary to draw the legend in all maps. Besides, authors have not written the units of the legend. The layout of this figure have to be improved. After that, do the same with the 26 maps of Figure 6. done
- Figure 4. Add the units of the legend. Done
- Figure 5 should be presented as Figure 4b because it is a zoom-in view of the map included in Figure 4. done
- How the MERRA2 databases were created? From which data? I cannot find the origin of this data. What is the meaning of MERRA2? Some aspects of this study are "dark" and authors should present all information in a "clearer" way.
https://gmao.gsfc.nasa.gov
MERRA-2 Data Access - Global Modeling and Assimilation ...
- Table 4. The content of this table can be presented as part of the text of section 3.4.2., and then, the table can be removed. done
- Most content of the Discussion section is a summary of the previous sections. I miss a deep analysis of the results on the basis of the existing literature. Without any doubt, this section has to be split into two sections: Discussion and Conclusions.
done
- References. The list is long and in some cases authors have not provided any detail/s from these documents. Therefore, I suggest two options: I) reduce the number of references to the most relevant ones; or II) add more information in the text from the cited literature. Please, avoid writing list of references without providing details; that is not valuable.
Ok. thank you so much. I will edit after final accept and I delete References that are not in the text.

Round 2
Reviewer 2 Report
Authors have partially addressed the comments that I made in the previous round of revision. Some aspects have not been solved, and appear as pending tasks. I encourage authors to refine and improve the quality of the text. These are the pending tasks:
- Authors have replied to one of my comments with this text "There are two deserts in the central plateau of iran Loot and central desert where dust storms are the main problems. Sand dunes are formed in these area. Wind erosion particlese are commonly transported by mutation and suspension. So there is no exact information on the particle size and i did not fond any research on this". This information about the presence of two deserts and the lack of available data in the literature HAVE to be included in the text of the manuscript."
- MERRA-2 Data. Authors HAVE to include the information about what is MERRA-2 and the link to this database in the text of the manuscript. Again, there is a lack of detail information about the data used in this study. Please, address this issue properly.
- Figures 4a and 4b have to be presented in the same figure, located at the left and right part, respectively. The current presentation is a mess.
- Figures 4a and 4b. I repeat the comment: ADD the units.
- Figure 6. The 24 maps have to be presented in one page, with 3 maps per row. Please, write the UNITS of the legend. The layout of this figure have to be improved.
- The numbering of figures is wrong in the revised manuscript. Please, check it throughout the text paying attention.
- Sub-sections "3.4.2. Linear regression" and "3.4.3. Nonlinear regression" have to be merged into a one sub-section called "3.4.2. Linear and nonlinear regression".
- References. The list is long and in some cases authors have not provided any detail/s from these documents. Therefore, I suggest two options: I) reduce the number of references to the most relevant ones; or II) add more information in the text from the cited literature. Please, avoid writing list of references without providing details; that is not valuable.
Author Response
Authors have replied to one of my comments with this text "There are two deserts in the central plateau of iran Loot and central desert where dust storms are the main problems. Sand dunes are formed in these area. Wind erosion particlese are commonly transported by mutation and suspension. So there is no exact information on the particle size and i did not fond any research on this". This information about the presence of two deserts and the lack of available data in the literature HAVE to be included in the text of the manuscript."
done
- MERRA-2 Data. Authors HAVE to include the information about what is MERRA-2 and the link to this database in the text of the manuscript. Again, there is a lack of detail information about the data used in this study. Please, address this issue properly.
done
- Figures 4a and 4b have to be presented in the same figure, located at the left and right part, respectively. The current presentation is a mess.
done
- Figures 4a and 4b. I repeat the comment: ADD the units.
done
- Figure 6. The 24 maps have to be presented in one page, with 3 maps per row. Please, write the UNITS of the legend. The layout of this figure have to be improved.
done
- The numbering of figures is wrong in the revised manuscript. Please, check it throughout the text paying attention.
done
- Sub-sections "3.4.2. Linear regression" and "3.4.3. Nonlinear regression" have to be merged into a one sub-section called "3.4.2. Linear and nonlinear regression".
done
- References. The list is long and in some cases authors have not provided any detail/s from these documents. Therefore, I suggest two options: I) reduce the number of references to the most relevant ones; or II) add more information in the text from the cited literature. Please, avoid writing list of references without providing details; that is not valuable.
done
This manuscript is a resubmission of an earlier submission. The following is a list of the peer review reports and author responses from that submission.